# Do Medical Graduates from a Rural Longitudinal Integrated Clerkship Work in Similar Rural Communities?

**DOI:** 10.3390/ijerph21121688

**Published:** 2024-12-18

**Authors:** Jessica Beattie, Lara Fuller, Marley J. Binder, Laura Gray, Vincent L. Versace, Gary D. Rogers

**Affiliations:** 1Rural Community Clinical School, School of Medicine, Deakin University, Colac, VIC 3250, Australia; lara.fuller@deakin.edu.au; 2Department of Rural Health, School of Medicine, Deakin University, Warrnambool, VIC 3280, Australia; m.binder@deakin.edu.au (M.J.B.); vincent.versace@deakin.edu.au (V.L.V.); 3The Damion DRAPAC Centre, School of Medicine, Deakin University, Geelong, VIC 3216, Australia; l.gray@deakin.edu.au; 4School of Medicine, Deakin University, Geelong, VIC 3216, Australia; g.rogers@deakin.edu.au

**Keywords:** Longitudinal Integrated Clerkships, rural medical education, rural health, general practice, primary care, medical workforce outcomes

## Abstract

(1) Background: Medical graduates who have undertaken longitudinal rural training have consistently been found to be more likely to become rural doctors and work in primary care settings. A limitation of such findings is the heterogeneous nature of rural medical education and contested views of what constitutes ‘rurality’, especially as it is often reported as a binary concept (rural compared to metropolitan). To address the identified gaps in workforce outcomes for rural medical training and to demonstrate accountability to the communities we serve, we investigated whether Longitudinal Integrated Clerkship (LIC) graduates are practicing in communities with similar rural classification to those where they trained. Within an LIC, students learn the curriculum in an integrated, simultaneous manner. (2) Material and Methods: A retrospective cohort study analysing variables associated with working in smaller rural communities. (3) Results: LIC graduates who undertook an additional year of rural training were five times more likely to work in communities of similar rurality to the program’s training footprint. (4) Conclusions: The duration of rural training alone did not lead to optimal rural workforce outcomes. However, graduates who had trained in a combination of rural settings, an LIC, and block rotation were the most likely to practice in communities of similar rurality to the clerkship’s training footprint. This highlights the impact of both the training duration and setting inclusive of an LIC on fostering positive rural workforce outcomes and the need to develop innovative solutions to expand these models of training in smaller rural communities.

## 1. Introduction

Internationally and within Australia, there is a well-documented and persistent maldistribution of the medical workforce [1,2,3]. This maldistribution is both geographic and vocational, culminating in compromised access to health care for rural communities, leading to poorer health outcomes compared to their urban counterparts [3,4,5]. Primary care is the foundation of healthcare delivery in rural communities and there is an enduring need to graduate more doctors who will pursue this specialty and, in particular, practice in smaller rural communities [1,5,6,7]. This aligns with universities’ social accountability obligation to ensure they graduate doctors who will meet the needs of the communities they serve [5,8]. One key strategy medical schools employ to meet this obligation is longitudinal rural medical training [5,7].

Medical graduates who have undertaken longitudinal rural training have consistently been found to be more likely to become rural doctors and work in primary care settings [9,10,11,12,13]. A limitation of such findings is the heterogeneous nature of rural medical education and contested views of what constitutes ‘rurality’, especially as it is often reported as a binary concept (rural compared to metropolitan) [14,15,16].

Rural medical education is primarily delivered by two prevailing clinical clerkship models: traditional ‘block rotations’ and Longitudinal Integrated Clerkships (LICs) [17,18]. Block rotations are the most common model and sequentially place the student in discipline-specific rotations, each for an intensive time-limited period (e.g., five weeks) [17]. In the rural context, a regional hospital is often the primary setting for this clerkship. Conversely, within LICs, students learn all the required disciplines in a simultaneous and integrated manner [17,18,19]. The structure of LICs allows the clerkship to be delivered in low-resource settings (such as smaller rural communities) via longitudinal attachments to both a rural primary care facility and a smaller rural hospital [18].

In recent years, many rural LICs have reached a level of maturity where universities can investigate their impact on rural workforce outcomes [15,20,21,22]. Research has found that when viewed alone or in comparison to metropolitan and rural block rotation training, rural LIC graduates are more likely to work rurally and in primary care specialties [15,20,21]. As the definition of what constitutes a rural community varies both internationally and within countries, a more granular analysis of the types of communities where LIC graduates work is required. The ubiquitous reporting of binary outcomes obfuscates the rich diversity in terms of geography, population demographics, economic conditions, and the bespoke healthcare needs of rural communities [7,10,16,23]. For example, within Australia, smaller rural communities are suffering from a significant disparity in the distribution of the medical workforce compared to large regional centres. Large regional centres have 109.9 full-time equivalent General Practitioners (GPs) per 100,000 population, compared to small rural towns which have 78.2 [24]. This means that residents of small rural towns have significantly reduced access to medical care, including general practice services. Specifically, residents of these communities’ access Medicare (Australia’s universal health insurance scheme) at a rate that is 50% lower than those living in metropolitan and larger regional areas [25,26].

In Australia, several co-existing factors are negatively affecting the current and future general practice (primary care) workforce: (i) a declining number of doctors choosing to specialize in general practice, (ii) a substantial number of GPs indicating impending retirement, and (iii) an ageing population, particularly in rural communities, increasing the demand on primary care services [5]. Collectively, these factors underscore the finite resources available to rural communities to train the next generation of doctors. Despite these challenges, these communities persist in their financial, social, and educational investment, hoping to encourage medical students to become rural clinicians [21,27]. An example of this investment is rural LIC supervisors, who, despite often simultaneously supervising multiple levels of learners as well as providing care for their patients, have reported that one of their primary motivations is to educate and recruit the next generation of rural clinicians, particularly for primary care [28,29,30]. To sustain this critical relationship between rural LIC supervisors and universities, we must evaluate and transparently report on the workforce outcomes of our program.

To address the identified gaps in workforce outcomes for longitudinal medical training and to demonstrate accountability, responsibility, and respect for the communities we serve, we investigated whether LIC graduates, compared to graduates on other clinical training pathways, are working in communities with similar rural classification to where they trained, as well as identifying the medical disciplines in which they are engaged. To address the latter point, our study only included graduates who were at least five years post-medical degree completion and therefore more likely to have completed their medical specialty training.

## 2. Materials and Methods

### 2.1. Setting

Deakin University’s Doctor of Medicine (MD) degree, formerly known as the Bachelor of Medicine/Bachelor of Surgery, began in 2008. This four-year graduate-entry program is delivered across both metropolitan and rural locations, primarily in western Victoria, Australia. In accordance with Australia’s Modified Monash Model (MMM) geographical remoteness and population classification system, the MD course is offered in various locations, including MM1 (metropolitan areas), MM2 (regional centres), MM3 (large rural towns), MM4 (medium rural towns), and MM5 (small rural towns) [31]. The MMM is a critical classification tool developed to help improve the equitable distribution of Australia’s health workforce in rural and remote areas and is used throughout this paper to explain rurality within our context [31].

The first two pre-clinical years, known as Foundations of Medicine, were based during the period of study in Geelong (MM1). In years three and four, students engaged in clinical training, referred to as Professional Practice of Medicine, at one of five clinical schools: Geelong or Eastern Health (MM1), Ballarat (MM2), Warrnambool (MM3), or the LIC known as the Rural Community Clinical School (RCCS) (MM3-5) (Figure 1).

The RCCS, a comprehensive LIC [18], began in 2010 and is located in the Southwest and Grampians regions of Victoria, Australia (Figure 1). On average, 20 students per year participate in the program. The duration of clinical training in year three extends throughout the entire academic year. Rural general practices affiliated with rural hospitals host LIC students, typically in groups of 2 to 4. Students engage in learning through parallel consulting sessions in general practice and participate in a variety of other clinical settings.

The RCCS locations are dispersed over a large geographic area, with the majority of the program’s geographic footprint classified as MM5 (small rural towns) [32]. The population of the rural towns where the students are located ranges from approximately 3500 to 20,000.

Following the LIC year, students complete their fourth year at either a regional/rural clinical school (MM2 or MM3) or a metropolitan clinical school (MM1) within a block rotation structure. Non-LIC students typically remain at their designated clinical school—whether metropolitan or rural—for the entire two years of clinical training.

### 2.2. Data Collection

Graduates’ 2023 Principal Place of Practice (PPP) were extracted from the Australian Health Practitioners Regulation Agency (AHPRA) register [33]. The PPPs were individually recorded by location, state, and postcode. Each location was geocoded and levels of rurality were assigned by the MM using a spatial join in ArcGIS [31].

To identify variables associated with graduates’ PPP and medical specialty, administrative data were extracted from the Deakin University School of Medicine graduate database. This data included gender, rural background, clinical school, bonded medical place (BMP) (see below), and graduation year. These variables were then matched with graduates’ AHPRA PPPs. The data were de-identified by a data manager and provided to researchers for analysis.

### 2.3. Data Analysis

PPP was primarily analysed by comparing metropolitan (MM1), regional (MM2), and rural (MM3-7) settings to address the research question of whether rural LIC graduates are working in communities with similar rural classification to those of the program’s training footprint. To provide a comparison for these data, all eligible MD graduates were included in this study. Individual categories of MM are also reported as aligned with recommendations for consistency in geographic reporting, providing evidence for meaningful comparison and use in future policy development [20,21].

Univariate associations between graduate characteristics, clinical school, medical specialty, and PPP were explored using Pearson’s chi-square tests. A *p*-value of ≤0.05 was considered statistically significant.

Multimodal regression was performed to determine odds ratios associated with working in regional and rural locations. All univariate associations with *p* ≤ 0.1 were retained and used in the multimodal modelling.

Graduates’ years of graduation were grouped into four categories by post-graduate year (PGY) for analysis to determine if there were any patterns associated with where graduates may be on the medical training continuum. Categories were PGY 5–6, PGY 7–8, PGY 9–10, and PGY 11–12.

Bonded Medical Places (BMPs) and Medical Rural Bonded Scholarships were combined for analysis as these schemes have since been amalgamated. BMPs are a return of service obligation that requires a graduate to work in a rural location for a specified amount of time, with the duration dependent on when the graduate entered the scheme and the practice location’s geographic level of rurality [34].

Graduates’ medical specialties were categorized into General Practitioner (GP), Non-GP specialists, and those with no recorded medical specialty. As rural communities require more doctors in primary care, known in Australia and from here on referred to as GPs (inclusive of rural generalists), an area of interest was the number and location of graduates working as GPs [5].

Training pathways were divided into four groups reflective of graduates’ clinical training experience: (i) metropolitan only at either Geelong or Eastern Health clinical schools (metro only), (ii) rural clinical school (RCS) block rotation at either Warrnambool or Ballarat (RCS), (iii) LIC and 4th-year metropolitan training (LIC/metro), and (iv) LIC and 4th year RCS training (LIC/RCS).

All analyses were undertaken in SPSS v27 (version 24, IBM, New York, NY, USA) [35].

### 2.4. Ethics Approval

Ethics approval was provided by the Deakin University Human Research Ethics Committee (2019-085).

## 3. Results

### 3.1. Participants

All Deakin graduates between PGY 5–12 (graduation years 2011–2018) were included in the study. There were 1041 graduates during this period, but after excluding cases without a registered Australian AHPRA PPP for 2023, 932 cases remained available for analysis.

### 3.2. Demographic Characteristics by Graduates’ Training Pathway

Table 1 provides demographic data for the entire sample by the graduates’ clinical training pathway. Overall, 50.5% were females, 28.4% had a BMP, 27.3% were from a rural background, and 13.4% had participated in the LIC. The majority of graduates, 81.3%, were working in metropolitan (MM1) locations. Of the 42% (n = 391) of graduates with a recorded medical specialty, general practice was the most common (61.9%, n = 242).

Notably, demographic data varied between clinical training pathways. Graduates of both training pathways that included the LIC were more likely to have a BMP (39.3% and 39%). The LIC/RCS pathway had the highest percentage of graduates with a rural background (41.5%). In the early years of the course (2011–2012), LIC graduates were more likely to remain rural for their 4th year (LIC/RCS) (n = 16) compared to later years (n = 7) (2017–2018) (*p* = 0.037).

### 3.3. Graduate Characteristics and Rurality of Work Location (2023)

The graduates’ characteristics were analysed by their PPP. In terms of working rurally, compared to regional or metropolitan locations, there was no significant difference based on the graduates’ PPP when compared to their gender or PGY year (Table 2). Significant variables were BMP (*p* = 0.012), rural background (*p* =< 0.001), training pathway (*p* =< 0.001), and medical specialty (*p* =< 0.001). The largest proportion of graduates working in rural communities were LIC/RCS graduates (26.8%), with a further 22% in regional communities. LIC/metro graduates were also working in rural communities in higher proportions (15.5%) when compared to regional communities (4.8%).

Working in rural communities in higher proportions compared to regional, respectively was also apparent for GPs (22.3% vs 10.7%), rural background graduates (18.5% vs 17.3%), and graduates with a BMP (14% vs 8.7%) (Table 2).

When multinomial regression was performed (reference group metropolitan training only), LIC/RCS graduates were 5.4 (95% CI 2.2–13.4) times more likely to work in rural communities when compared to graduates who had only undertaken metropolitan training (Table 3). This group were also 4.2 (95% CI 1.7–10.1) times more likely to work in regional communities. Notably, LIC/metro graduates were twice as likely to work in rural communities compared to graduates with metropolitan training only (95% CI 0.97–4.4). Graduates who had attended a rural block rotation for 2 years (RCS group) were 1.5 and 1.4 times more likely, respectively, to work regionally or rurally.

### 3.4. Medical Specialty and Geographic Workforce Location

No statistical significance was found between graduates’ training pathways and their registered medical specialty (Table 1). A comparison of the graduates found that at this time point (2023), all training pathways produced more GPs than other medical specialties. Briefly, GPs represented the following percentages of those with a registered vocational qualification for each training pathway: 75% for LIC/RCS, 72% for LIC/metro, 65.3% for RCS, and 57.9% for metro only (Table 1).

GPs were more likely to be working rurally compared to all other graduates (*p* =< 0.001). Notably, 22.3% of GPs worked in rural communities, compared to 0% of non-GP specialists and 6.3% of graduates without a registered medical specialty. GPs were working in regional locations at slightly higher percentages (10.7%), compared to no recorded medical specialty (9.2%) and non-GP specialists (6.7%).

There were statistically significant differences in the work location of registered GPs based on their training pathways (*p* =< 0.001). For both LIC training pathways, over 50% of their GPs worked in rural communities. LIC/RCS GP graduates were working in regional locations in the highest proportions (26.7%), while, interestingly, no LIC/metro graduates were working in regional locations (Figure 2).

## 4. Discussion

This study found that the duration of rural training alone does not guarantee optimal rural workforce outcomes. Instead, the combination of rural clinical settings—LIC and traditional block rotations—yielded the highest odds ratios for regional (4.2) and rural (5.4) work. In contrast, graduates who completed two years of a traditional rural block rotation (RCS) were only 1.5 and 1.4 times more likely to work regionally or rurally, respectively, compared to those who trained solely in metropolitan areas. This finding adds to the literature on rural workforce outcomes by challenging the conventional view of longitudinal rural medical training as a uniform experience, suggesting that it should be reported in a more nuanced way. It highlights that both the duration of training (at least two years) and a clinical training experience that includes a rural LIC can significantly influence workforce outcomes [36].

A potential explanation for this phenomenon is that training students across various rural clinical settings inclusive of general practice and providing a positive training experience may enhance what has been termed rural self-efficacy: in essence, graduates’ belief that they have the skills for rural practice [37,38]. This combination of training is arguably more aligned with how modern healthcare delivery has evolved, from less reliance on in-patient care to more care delivered in community settings [39,40]. LIC/RCS graduates were exposed to multiple medical disciplines continuously across various clinical settings, arguably providing a more holistic and realistic understanding of rural medicine. In previous work, LIC graduates who became rural GPs noted that their career decisions were shaped by various contextual factors embedded in the clerkship. These included the development of long-term, trusting relationships with rural supervisors, whom they viewed as role models; a sense of comfort and compatibility in the general practice environment; and exposure to the diversity and breadth of rural general practice [41].

Notably, the LIC/metro pathway was associated with a higher odds ratio for working rurally compared to the two-year RCS pathway, despite graduates on the former pathway completing their final year in a metropolitan setting. This suggests rural LICs may have a lasting impact on workforce outcomes, even when this was students’ only year of rural training. Additionally, LIC/metro graduates were more likely to work in smaller rural communities that resembled the LIC’s training footprint than regional communities. To gain a deeper understanding of LIC’s influence, a comparative analysis of a two-year LIC program against a combined LIC/RCS pathway would be valuable.

The demographic differences between training pathways warrant consideration as an additional explanation for our findings. LIC/RCS graduates were more likely to come from a rural background and both LIC training pathways had a higher proportion of graduates with a return of service obligation (BMP). This suggests that students with a pre-existing intention to become rural doctors may be self-selecting to participate in the LIC program. However, in isolation, this does not adequately explain the LIC/RCS enhanced rural work outcomes, as all rural training pathways had more rural background and return of service graduates compared to metropolitan training only. To address the notion of self-selection, future studies should consider measuring students’ clinical school preferences against their workforce outcomes.

The results reinforce the significance of the clinical training pathway in redressing the maldistribution of the medical workforce and underscore the importance of investing in and enhancing the capacity to train medical students outside metropolitan and regional centres [25]. LICs have been extensively validated as a clinical clerkship model that can be effectively delivered in resource-limited communities [18]. However, our data indicate that only a small proportion of our graduates had the opportunity to participate in the LIC/RCS pathway, which limits the potential for significant change in the rural medical landscape. Innovation and ‘flipped’ models of training (where students are based in rural communities and, if required, rotate to larger regional or metropolitan communities for key experiences) can facilitate further rural and remote communities to be the primary base of longitudinal rural medical education [7]. This will allow more medical graduates to be trained in and for rural communities and aligns with the growing concept of place-based education. Place-based education prioritizes experiential, community-based, and contextual learning, is responsive to the needs of the communities, and therefore creates a greater connection between the learner and the community [42].

While building capacity to train more graduates in rural communities is one solution to increase the number of LIC graduates, this goal is hindered by supervision constraints relating to the existing maldistribution of the medical workforce and will require time and innovation to readdress. A more immediate change is to prioritize selecting the right individuals for training in rural communities through the strategic alignment of known influential factors such as rural LIC participation and candidates who have a rural background. Our previous workforce tracking employed the binary definition of rurality and found that 66.7% of graduates with a combination of rural background and LIC/RCS training were working regionally or rurally [10]. In this vein, we have implemented a dedicated Rural Training Stream (2022) that integrates both elements, but with a place-based focus prioritizing recruiting students from our rural footprint to train there. The students within this training stream who undertake the LIC will also stay within the rural footprint for their final year [43]. Over time this will increase the number of students who follow the LIC/RCS pathway.

A further strength of this study compared to our previous work was that graduates have progressed along the medical training continuum and many now have a registered medical specialty, most commonly general practice. While the specialty distribution did not show statistical variation between training pathways, this will remain an area of ongoing focus as more graduates complete their vocational training, recognizing that general practice typically requires a shorter duration than some other specialties. Encouragingly, for both LIC training pathways, at least half of the graduates working as GPs were working in communities with a similar rural classification to the clerkship’s training footprint.

As we build on our strategic direction to ensure we are actively implementing evidence-based policies to meet our social accountability obligation, further graduate tracking work will be undertaken reporting on the workforce outcomes for what has become our formally defined rural training footprint [32]. Furthermore, as we now have a longitudinal repository of graduates’ work locations, the geographic migration patterns of LIC graduates will be investigated to determine who returns rural and who stays rural. This is a gap in the literature as most studies, including this one, only report on a single time point [15].

This study had several limitations. It relied on the PPP recorded by AHPRA. Despite this being authenticated as a valid graduate tracking method, it does not provide information on other locations where graduates may be working [44]. For example, many metropolitan-based non-GP medical specialists provide services to regional and rural communities periodically. Hence, our finding that no non-GP specialist had a PPP in rural communities may not be reflective of an absence of work in these communities. A further limitation is that the number of graduates who have undertaken the LIC is relatively small, particularly those on the LIC/RCS pathway, and therefore the results should be interpreted with caution and may not be transferrable to other rural LIC programs. It is anticipated that the number of LIC/RCS graduates will increase in the coming years with the introduction of the RTS.

## 5. Conclusions

The duration of rural training alone did not lead to optimal rural workforce outcomes. However, graduates who had participated in the rural LIC and completed a second year of rural training were the most likely to practice in communities of similar rurality to the LIC training footprint. Moreover, both LIC training pathways produced the highest proportion of graduates who were working in rural communities as GPs, demonstrating the clerkship’s effectiveness at addressing both geographic and specialty maldistribution. Findings reinforce the necessity to efficiently manage existing rural medical training resources and develop innovative solutions to expand training capacity. By increasing the number of LIC students who remain rural throughout their clinical training, we can improve workforce outcomes. This targeted strategy is essential for addressing the persistent challenges of healthcare access and workforce distribution in rural settings.

## Figures and Tables

**Figure 1 ijerph-21-01688-f001:**
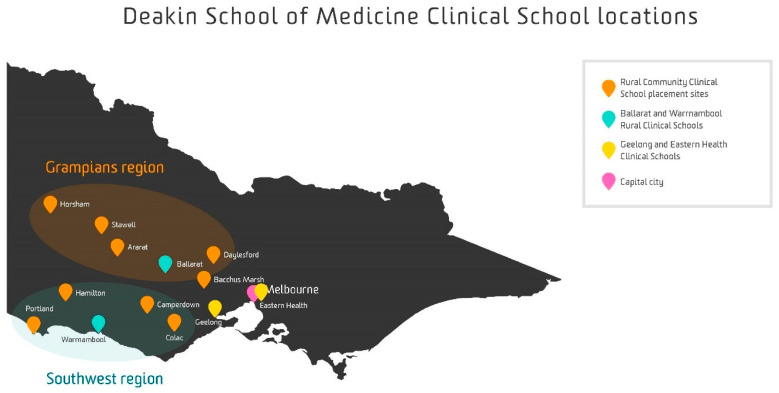
Deakin University School of Medicine clinical school locations. Reproduced with permission.

**Figure 2 ijerph-21-01688-f002:**
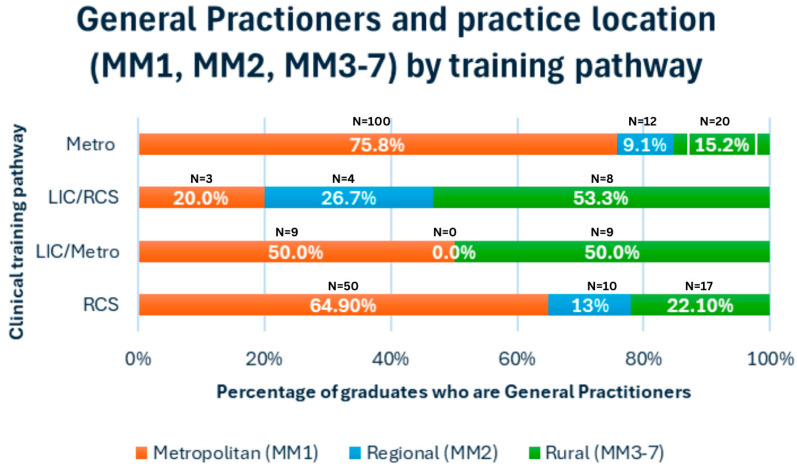
Geographic work location of General Practitioners (n = 242), 2023 (metropolitan (MM1), regional (MM2), and rural (MM3-7)), by graduate training pathway.

**Table 1 ijerph-21-01688-t001:** Descriptive data on all Deakin University medical graduates with a 2023 Australian Principal Place of Practice (2011–2018) and associations between cohort characteristics of clinical training pathways.

Variable	All Graduates (%, N)	Metro Only (%, N)	Rural Clinical School (%, N)	LIC and Metro 4th Year (%, N)	LIC and RCS 4th Year (%, N)	*p* Value
**All graduates**	932	531 (57%)	276 (29.6%)	84 (9%)	41 (4.4%)	
**Gender**						
Male	49.5 (461)	49.0 (260)	52.9 (146)	51.2 (43)	53.7 (22)	0.720
Female	50.5 (471)	51.0 (271)	47.1 (130)	48.8 (41)	46.3 (19)	
**Return of Service Obligation (BMP)**						
BMP	28.4 (265)	24.7 (131)	30.8 (85)	39.3 (33)	39.0 (16)	0.009 *
No BMP	71.6 (667)	75.3 (400)	69.2 (191)	60.7 (51)	61.0 (25)	
**Rural background**						
Yes	27.3 (254)	23.9 (127)	30.4 (84)	31.0 (26)	41.5 (17)	0.027 *
No	72.7 (678)	76.1 (404)	69.6 (192)	69.0 (58)	58.5 (24)	
**Postgraduate year (PGY)**						
PGY 5–6	24.1 (225)	22.8 (121)	25.0 (69)	33.3 (28)	17.1 (7)	0.037 *
PGY 7–8	26.3 (245)	24.3 (129)	26.4 (73)	33.3 (28)	22.0 (9)	
PGY 9–10	25.6 (239)	27.9 (148)	25.4 (70)	21.4 (18)	22.0 (9)	
PGY 11–12	23.9 (223)	25.0 (133)	23.2 (64)	11.9 (10)	39.0 (16)	
**Medical specialty**						
None recorded	58.0 (541)	57.1 (303)	57.2 (158)	70.2 (59)	51.2 (21)	0.094
General Practice/Rural Generalist	26.0 (242)	24.9 (132)	27.9 (77)	21.4 (18)	36.6 (15)	
Non-GP specialties	16.0 (149)	18.1 (96)	14.9 (41)	8.3 (7)	12.2 (5)	
**Principal Place of Practice (2023) Modified Monash (MM)**						
MM1	81.3 (758)	85.7 (455)	77.9 (215)	79.8 (67)	51.2 (21)	<0.001 *
MM2	9.2 (86)	7.7 (41)	11.6 (32)	4.8 (4)	22.0 (9)	
MM3	2.8 (26)	1.9 (10)	3.6 (10)	3.6 (3)	7.3 (3)	
MM4	3.3 (31)	2.8 (15)	2.2 (6)	6.0 (5)	12.2 (5)	
MM5	2.6 (24)	1.5 (8)	4.0 (11)	3.6 (3)	4.9 (2)	
MM6	0.3 (3)	0.2 (1)	0.7 (2)	0.0	0.0	
MM7	0.4 (4)	0.2 (1)	0.0	2.4 (2)	2.4 (1)	

* Pearson Chi-squared test. The *p*-value of ≤0.05 is considered significant.

**Table 2 ijerph-21-01688-t002:** Association between cohort characteristics and Principal Place of Practice: metropolitan (MM1), regional (MM2), and rural (MM3-7).

Variable N	MM1 Percentage (%, N)	MM2 Percentage (%, N)	MM3-7 Percentage (%, N)	*p* Value
All graduates (n = 932)	81.4% (758)	9.2% (86)	9.4% (88)	
**Gender**				
Female (n = 461)	82.0 (378)	8.7 (40)	9.3 (43)	0.830
Male (n = 471)	80.7 (380)	9.8 (46)	9.6 (45)	
**BMP**				
Yes (n = 265)	77.4 (205)	8.7 (23)	14.0 (37)	0.012 *
No (n = 667)	82.9 (553)	9.4 (63)	7.6 (51)	
**Rural background**				
Yes (n = 254)	64.2 (163)	17.3 (44)	18.5 (47)	<0.001 *
No (n = 678)	87.8 (595)	6.2 (42)	6.0 (41)	
**Training pathway**				
Metro only (n = 531)	85.7 (455)	7.7 (41)	6.6 (35)	<0.001 *
RCS (n = 276)	77.9 (215)	11.6 (32)	10.5 (29)	
LIC/metro (n = 84)	79.8 (67)	4.8 (4)	15.5 (13)	
LIC/RCS (n = 41)	51.2 (21)	22.0 (9)	26.8 (11)	
**Postgraduate year (PGY)**				
PGY 5–6	83.6 (188)	8.4 (19)	8.0 (18)	0.340
PGY 7–8	77.8 (186)	12.1 (29)	10.0 (24)	
PGY 9–10	83.7 (205)	8.6 (21)	7.8 (19)	
PGY 11–12	80.3 (179)	7.6 (17)	12.1 (27)	
**Medical specialty**				
None recorded (n = 5 41)	84.5 (457)	9.2 (50)	6.3 (34)	<0.001 *
GP (n = 242)	66.9 (162)	10.7 (26)	22.3 (54)	
Non-GP specialist (n = 149)	93.3 (139)	6.7 (10)	0.0 (0)	

* Pearson Chi-squared test. A *p*-value of <0.05 is considered significant.

**Table 3 ijerph-21-01688-t003:** Multinomial regression and odds ratio of graduates who were working in regional (MM2) and rural (MM3-7) communities.

	MM2 Odds Ratio (CI 95%)	MM3-7 Odds Ratio (CI 95%)
Metro background (parameter)	1	1
Rural background	3.7 (CI 2.3–5.9)	3.8 (2.3–6.1)
No BMP (parameter)	1	1
BMP	0.8 (0.5–1.4)	1.5 (0.9–2.5)
Metro training only (parameter)	1	1
RCS	1.5 (0.9–2.5)	1.4 (0.8–2.5)
LIC/Metro	0.6 (0.2–1.8)	2.0 (0.9–4.4)
LIC/RCS	4.2 (1.7–10.1)	5.4 (2.2–13.4)
Medical specialty		
None recorded (parameter)	1	1
GP	1.4 (0.8–2.4)	4.4 (2.7–7.2)
Non-GP spec	0.6 (0.3–1.3)	NA (0 cases)

Retained variables for the model from Table 2 that were *p* =< 0.1. Reference group MM1.

## Data Availability

Data are available upon reasonable request by contacting Jessica Beattie.

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
