# Peer review of "Do Medical Graduates from a Rural Longitudinal Integrated Clerkship Work in Similar Rural Communities?"

_ijerph, 2024, doi:10.3390/ijerph21121688_

Round 1

Reviewer 1 Report

Comments and Suggestions for Authors

Thankyou for the opportunity to review your manuscript. Your manuscript addresses an important gap in knowledge. I have a number of suggestions for your consideration.

Abstract: could you very briefly describe LIC for an audience who may not be familiar with the term, and demonstrate the link between LIC and the values highlighted in lines 18-20?

Would it be useful to be consistent with the term Eastern Health or Box Hill, rather than use both terms?

Discussion:

Could an alternate explanation be around holistic or more true to life experiences (Lines 256-259), or the option of multiple career pathways?

Could you also comment on the value of increasing provision of LIC in increasingly rural and remote settings?

Some minor points: could you please check the correct tense is used in the abstract and in the introduction? It is useful to refer to the graduates as 'our graduates' or would it be better to remain removed?

Is it unusual to use a significant cutoff of less than or equal to 0.05, or 0.01?

Could you please check consistency with reporting of decimal points, eg in tables?

Author Response

Reviewer 1. 

Thankyou for the opportunity to review your manuscript. Your manuscript addresses an important gap in knowledge. I have a number of suggestions for your consideration.

Comment: Abstract: could you very briefly describe LIC for an audience who may not be familiar with the term, and demonstrate the link between LIC and the values highlighted in lines 18-20? Response: A brief explanation of an LIC has been included. To meet the word limit we have removed some of the words from lines 18-20 (they remain in the main text).

Comment: Would it be useful to be consistent with the term Eastern Health or Box Hill, rather than use both terms? Response: Thank you this has been changed to Eastern Health to align with Figure 1.

Discussion:

Comment: Could an alternate explanation be around holistic or more true to life experiences (Lines 256-259), or the option of multiple career pathways? Response: Thank you. This is a helpful observation, and we have added some further information highlighting that training across rural settings provides continuous exposure to medical disciplines in different settings therefore giving the student more realistic exposure to rural medicine.

Comment: Could you also comment on the value of increasing provision of LIC in increasingly rural and remote settings? Response: Thank you. We have expanded on this in the discussion describing the value of and linking with place-based learning.

Comment: Some minor points: could you please check the correct tense is used in the abstract and in the introduction? It is useful to refer to the graduates as 'our graduates' or would it be better to remain removed? Response: Thank you ‘our’ has been removed.

Comment: Is it unusual to use a significant cutoff of less than or equal to 0.05, or 0.01? Response: A p-value of equal or less than 0.05 is commonly statistically significant. If this comment is referring to the retainment of values of 0.1 (not 0.01) for the multimodal analysis this is also a valid and commonly used.

Comment: Could you please check consistency with reporting of decimal points, eg in tables? Response: Thank you. This has been addressed

Reviewer 2 Report

Comments and Suggestions for Authors

It is a good paper, with some minor corrections it can be published. It is a very technical paper with relevance for the Australian context and not easy to be understood by readers from  other countries.

My concerns are:

1. Research question was not clearly addressed.

2. It was not very clear expressed about new insights of the paper.

3. Authors published already a couple of papers on the same subject and there referred themselves in the paper.

4. The title is very long and not very clear and they used abbreviations in the abstract and they did not explained them.

Author Response

Comment: It is a good paper, with some minor corrections it can be published. It is a very technical paper with relevance for the Australian context and not easy to be understood by readers from other countries. Response: Thank you. We have added some additional information about the geographic remoteness and population scale (under setting) which we hope will enhance understanding.

Comment: My concerns are:

  1. Research question was not clearly Response: Thank you. Our research question was as stated to investigate whether our LIC graduates, compared to graduates on other clinical training pathways, are working in communities with similar rural classification to where they trained, as well as identify the medical disciplines in which they are engaged. Our results demonstrated that LIC/RCS students were the most likely group to work in smaller rural communities (similar to the LIC training footprint) and that for both LIC training pathways approximately half of the graduates who were GPs were working in ‘rural’ communities. Also LIC/metro graduates were more likely to work rurally, despite only one year of rural training compared to RCS graduates. We would be happy to address this further but would like further clarification.
  2. Comment: It was not very clear expressed about new insights of the Response:A gap in the literature has been, as described the homogeneous reporting of longitudinal rural training and the need to separate it by the types of training that are offered. Furthermore, as highlighted in the scoping review we undertook, there is a need for a more granular analysis of rurality and aligning our analysis to determine if the LIC graduates are more likely to work in similar communities to the training footprint. The linking of the medical specialty to what smaller rural communities need, General Practitioners (inclusive of Rural Generalists) is a new insight and will act as an important baseline to follow in the future. This paper only included graduates who were postgraduate years 5-12 as we deemed they were more likely to have completed vocational training. Also breaking down the clinical training pathways by demographic data is not commonly done so this added new insight, with rural bonding and rural background alone arguably not convincing explanations for the LIC/RCS outcomes. We have added some further information in the discussion around the influence of the LIC from both pathways.
  3. Comment: Authors published already a couple of papers on the same subject and there referred themselves in the paper. Response: Thank you. We have not published on the same subject but have published on the same subject area (rural LICs). This paper will form part of a PhD and was conceived from gaps and recommendations from a scoping review the lead author undertook that found that future workforce tracking of LIC graduates should include if they are working in communities with a similar geographic classification.
  4. Comment: The title is very long and not very clear and they used abbreviations in the abstract and they did not explained them. Response: The abbreviations have been addressed in the abstract. Aligned with this suggestion we have changed the title to ‘Do medical graduates from a rural LIC work in similar rural communities?’

Reviewer 3 Report

Comments and Suggestions for Authors

I found your article interesting and agree it constitutes a modest empirical contribution to the relevant literature. That said, I can offer a few observations for your consideration. First, I am not sure a community of 20-25,000 can be considered "rural" though perhaps Australian law so defines it? I found that assertion strange in any case. Second, while you acknowledge on page 7 that your study raises the question of self- selection (very real, I think), you do not suggest how this concern might b overcome with future work. Perhaps you could identify a sample experiencing the different forms of training/education and ask those individuals what their proclivities concerning where to practice and why are?  Second, and related, you suggest rural background seems to be important factor in decisions to practice in such communities and it may well be but I would like to suggest that you urge additional consideration of this matter in two ways. First, as just noted and second, by exploring any differences in training apart from length of time in community between the models of interest that might seem material. Maybe the character of their educations/rotations do not matter or otherwise identical apart from length of time, but I found myself curious on this question.  Third, as you note, your sample was small. Any way to obtain a larger one across time? Fourth, in line 44, "school" should be plural.

Author Response

Comment: I found your article interesting and agree it constitutes a modest empirical contribution to the relevant literature. That said, I can offer a few observations for your consideration.

Comment: First, I am not sure a community of 20-25,000 can be considered "rural" though perhaps Australian law so defines it? I found that assertion strange in any case. Response: Thank you. We agree there are variations in what is termed rural. Within Australia, the Modified Monash Model remoteness and population scale was developed specifically to help distribute the health workforce more equitably in rural and remote areas. This measure is referenced but here is the link Modified Monash Model | Australian Government Department of Health and Aged Care. We have included some additional information explaining that the scale relates to health workforce distribution. Here is some further background information; Commonly regional towns with up to 100,000 population are included in the binary analysis of rural so this paper sought to remove larger (regional) communities. Figure 1 illustrates where the LIC towns are located. Horsham is the only MM3 town and has the largest (20,000) population, but this town is also arguably one of the ‘remotest’ towns where our students train,  due to where it is geographically located (in what is referred to as the Wimmera Southern Mallee subregion which is dryland farming) and has a persistent medical workforce shortage.

Comment: Second, while you acknowledge on page 7 that your study raises the question of self- selection (very real, I think), you do not suggest how this concern might be overcome with future work. Perhaps you could identify a sample experiencing the different forms of training/education and ask those individuals what their proclivities concerning where to practice and why are?  Response: Thank you for your suggestion. We have included some further information about how self-selection could be compared to workforce outcomes in the future. We agree this is an area of future focus.

Comment: Second, and related, you suggest rural background seems to be important factor in decisions to practice in such communities and it may well be but I would like to suggest that you urge additional consideration of this matter in two ways. First, as just noted and second, by exploring any differences in training apart from length of time in community between the models of interest that might seem material. Maybe the character of their educations/rotations do not matter or otherwise identical apart from length of time, but I found myself curious on this question.  Response: Thank you for your comment. Rural background has consistently been found to be associated with fostering positive rural workforce outcomes. For clarification is the suggestion we explore the difference in training pathways by rural background? If so we agree this is of significant interest and it is an area we are seeking to explore further once our Rural Training Stream students graduate. Currently, a limitation to undertaking this work would be the sample size of rural background students within each training pathway. Or is this comment about a more nuanced investigation of contextual elements of the clerkship? If it’s the latter we have completed a grounded theory study on how a rural LIC influences graduates career decisions. It is referenced.

Comment: Third, as you note, your sample was small. Any way to obtain a larger one across time? Response: This is addressed in the discussion which explains that LIC students who are part of the dedicated Rural Training Stream will now stay rural for their fourth year, hence increasing the numbers of LIC/RCS graduates in the future. Each year the sample size will grow. Also in the limitations this is now further explained.

Comment: Fourth, in line 44, "school" should be plural. Response: Thank you. This has been fixed.